# Dietary Influences on the Longevity and Reproductive Success of *Anopheles aquasalis* in Laboratory Studies: Sucrose vs. Honey

**DOI:** 10.3390/insects15120978

**Published:** 2024-12-10

**Authors:** Fernanda Oliveira Rezende, Dimas Augusto da Silva, Sara Comini, Silvana de Mendonça, Ellen Santos, Lívia Baldon, Bruno Marçal, Amanda Cupertino de Freitas, Rafaela Moreira, Viviane Sousa, Mariana Lima, Marcele Rocha, Luciano A. Moreira, Alvaro Ferreira

**Affiliations:** 1Mosquitos Vetores: Endossimbiontes e Interação Patógeno-Vetor, Instituto René Rachou-Fiocruz, Belo Horizonte 30190-002, Brazil; fernanda.rezende@fiocruz.br (F.O.R.); dimas.silva@fiocruz.br (D.A.d.S.); saragrangeiro@live.com (S.C.); smendonca@aluno.fiocruz.br (S.d.M.); livia.baldon@aluno.fiocruz.br (L.B.); bmarcal@aluno.fiocruz.br (B.M.); acfreitas@aluno.fiocruz.br (A.C.d.F.); rafaela.moreira@fiocruz.br (R.M.); viviane.pauline@fiocruz.br (V.S.); alveslima.mariana@gmail.com (M.L.); marcelebio@yahoo.com.br (M.R.); luciano.andrade@fiocruz.br (L.A.M.); 2Departamento de Bioquímica e Imunologia, Instituto de Ciências Biológicas, Universidade Federal de Minas Gerais, 6627-Pampulha, Belo Horizonte 31270-901, Brazil; ellen.caroline24@gmail.com; 3Laboratório de Ecologia do Adoecimento & Florestas NUPEB/ICEB, Universidade Federal de Ouro Preto, Ouro Preto 35402-163, Brazil

**Keywords:** *Anopheles aquasalis*, malaria, vector–parasite interactions, adult mosquito diet, laboratory studies

## Abstract

Malaria is a serious health problem in many tropical and subtropical regions. *Anopheles aquasalis* mosquitoes are important in malaria research because they help us understand how malaria spreads. These mosquitoes, like other insects, have different needs and behaviors that affect their survival and ability to reproduce. One aspect that has not been well studied is how their diet influences these factors. In our research, we tested two types of food for adult mosquitoes: sucrose (a type of sugar) and honey. We wanted to see how these diets affected the mosquitoes’ lifespan, ability to reproduce, and the success rate of their eggs. Our results showed that mosquitoes fed honey lived longer and produced more eggs compared to those fed sucrose. The eggs from honey-fed mosquitoes also hatched more successfully. These findings are important because they show that the type of food mosquitoes eat can significantly impact their reproduction and survival. Understanding these effects can help improve laboratory studies on how malaria is spread and how we might better control it.

## 1. Introduction

Malaria, a severe infectious disease, has profoundly impacted human health for centuries and remains a major global health threat [1,2,3]. This disease is caused by protozoan parasites of the *Plasmodium* genus, with *Plasmodium falciparum* and *Plasmodium vivax* being the primary contributors to the public health burden [1,2,3,4]. Transmission occurs when female mosquitoes of the *Anopheles* genus, infected with *Plasmodium* spp., bite humans [1,2,3,4]. The *Plasmodium* sporozoites enter the bloodstream and migrate to the liver, where they replicate within hepatocytes, forming merozoites that are released into the blood [1,2,3,4,5]. Some of these blood-stage parasites develop into sexual forms known as gametocytes, which are taken up by another mosquito during a bite [5]. Inside the mosquito, gametocytes undergo fertilization and sporogonic development in the midgut, producing infectious sporozoites that migrate to the salivary glands, ready to infect a new host [5].

Although both *Anopheles aquasalis* and *Anopheles darlingi* are susceptible to *Plasmodium vivax* infection, *Anopheles aquasalis* is recognized as the primary vector in the coastal regions of Central and South America [6,7,8]. Due to its significant public health importance and the ease of laboratory rearing, *Anopheles aquasalis* has been extensively used as a neotropical anopheline model to study host–parasite interactions, with a particular focus on vector competence [6,9,10]. Vector competence refers to the physiological ability of a mosquito to become infected with and transmit pathogens, and it is influenced by both genetic and environmental factors [11]. Numerous laboratory studies have investigated the vector competence of *Anopheles*, aiming to understand the genetic mechanisms underlying immune responses to malaria parasites [12,13]. While these studies have uncovered key mechanisms involved in preventing or limiting parasite development, increasing evidence shows that abiotic and biotic factors, as well as within-vector environmental conditions, critically influence vector–parasite interactions [14,15]. Factors such as temperature, mosquito diet, insecticide exposure, microbial gut flora, infection history, and mosquito age [16,17,18,19,20,21,22,23,24,25] all play a role in modulating these interactions. Importantly, the availability and quality of food resources are crucial environmental variables that affect host–parasite relationships, influencing host susceptibility, parasite infectivity, virulence, and overall disease dynamics [18,26]. In addition to their blood-feeding behavior, female mosquitoes consume sugars from sources such as floral and extra-floral nectaries, fruits, and phloem sap, which have significant biological implications [27,28,29,30]. These non-blood dietary components provide essential sugars that are crucial for various biological processes, including the synthesis of amino acids and nucleotides, and the production of storage molecules like triglycerides, trehalose, and glycogen [27,30]. Recent research has demonstrated that the specific plant nectars consumed by *Anopheles* mosquitoes can influence their microbiome composition, which in turn affects their susceptibility to *Plasmodium* infection [27,30,31,32,33]. However, the impact of different sugar diets used in laboratory settings on the fitness traits of *Anopheles aquasalis*, such as survival and reproductive parameters, remains largely unexplored.

The specific effects of sucrose solutions, commonly used in laboratory colonies, on the survival and reproductive fitness of mosquitoes, particularly *Anopheles aquasalis*, remain unclear. To address this, we evaluated the impact of two different adult diets—sucrose and honey—on the longevity, fertility, and fecundity of *Anopheles aquasalis*. A 10% sucrose solution was chosen as it is the standard diet used in most laboratory studies, providing a baseline for comparison. In contrast, honey, which contains additional nutrients found in the natural diet of mosquitoes, was tested to assess if these nutrients could result in fitness differences. The primary focus of this study was to evaluate whether feeding mosquitoes a sugar solution composed solely of sucrose could mimic the natural feeding composition (nectar and pollen) encountered in the wild and to analyze the solution’s effects on the longevity and fecundity of *Anopheles aquasalis*. As such, the study aimed to guide laboratory research on mosquito feeding solutions and to contribute to the development of feeding protocols that better resemble natural feeding conditions. Our observations revealed that the type of sugar diet significantly affects mosquito survival as well as both fecundity and fertility rates. These findings highlight the substantial influence of diet on the reproductive fitness of *Anopheles aquasalis*, underscoring its potential implications for laboratory studies on vector–parasite interactions.

## 2. Materials and Methods

### 2.1. Mosquito Lineages and Mosquito Rearing

All mosquitoes used in this study were from a laboratory population established in 1995, derived from wild-caught adults collected in the municipality of Guapimirim, Rio de Janeiro state, Brazil. The rearing of *Anopheles aquasalis* mosquitoes was carried out following a protocol previously described in Rezende 2013. Briefly, larvae were reared in plastic containers measuring 40 cm × 12 cm × 4 cm, at a density of 100 to 120 larvae per liter, in a solution containing 0.33% salt (sodium chloride), resembling the salinity found in the natural habitat of this mosquito species. Whenever the saline solution appeared cloudy, it was replaced with a fresh one. The larvae were fed with crushed and sieved TetraMin Flakes. The pupae were collected and transferred to plastic cups containing a 0.33% saline solution and placed in cages. After emerging, adults were kept in 30 cm × 30 cm × 30 cm BugDorm insect cages, where mosquitoes were fed with 10% sucrose solution or 10% honey solution *ad libitum*. All mosquitoes were reared under insectarium-controlled conditions, at 28 °C and 70–80% relative humidity, in a 12/12 h light/dark cycle.

### 2.2. Nutritional Regimes

After they emerged from pupae, adult mosquitoes were provided with either a 10% sucrose solution or 10% honey solution, both *ad libitum*. Solutions were replaced three times a week. Seven days after emerging, adult females were exposed to a mouse (Swiss mice lineage—FIOCRUZ CEUA license LW 10/2026) for a blood meal, for 30 min. For egg-laying, plastic cups with a solution containing 0.33% salt and a filter were placed inside the cages. To address the ethical considerations regarding blood feeding on Swiss mice, we implemented a humane anesthesia protocol to ensure that the animals did not suffer during the procedure. We administered a combination of ketamine (100 mg/kg) and xylazine (10 mg/kg) via intraperitoneal injection. This approach was selected to maintain an adequate depth of anesthesia, ensuring the mice were comfortable and unresponsive to pain during the blood-feeding events.

Additionally, we set up the blood-feeding process so that no more than 20 mosquitoes fed on each mouse, guaranteeing that no more than 10% of the mouse’s blood volume was extracted. This measure was crucial in minimizing stress and ensuring the well-being of the mice. We closely monitored the mice throughout the procedure, assessing their vital signs and ensuring they remained stable. Post-anesthesia, the mice were kept warm to facilitate recovery, and we provided analgesia, such as buprenorphine, if the blood-feeding process extended beyond a reasonable duration. These measures reflect our commitment to ethical practices and compliance with animal welfare guidelines in research.

### 2.3. Survival Assays

Survival analyses were conducted on both female and male *Anopheles aquasalis* mosquitoes assigned to one of two adult nutritional regimes: either a 10% sucrose solution or a 10% honey solution provided *ad libitum*. After sex-sorting, 30 female pupae and 30 male pupae were transferred to plastic cups with a solution containing 0.33% salt and placed in paperboard cages (10 cm × 15 cm diameter), with ab libitum access to either a 10% sucrose solution or a 10% honey solution. Females and males were kept in the same cage during the entire course of the experiment. Seven days after emerging, adult females were exposed to a mouse (Swiss mice lineage—FIOCRUZ CEUA license LW 10/2026) for a blood meal, for 30 min. Two days after the blood meal, plastic cups with a solution containing 0.33% salt and a filter were placed inside the cages for egg-laying. Seven days after the first blood meal, another blood meal was administered, and this procedure was repeated until all female mosquitoes had deceased (a total of 7 blood meals were performed). The mortality of mosquitoes (females and males) was checked daily, and longevity was recorded as the number of days from emergence to death (hereafter referred to as longevity). Three replicates were performed for each experimental condition to ensure the robustness of the results.

### 2.4. Fecundity and Fertility Assays

For fecundity and fertility analyses, female and male *Anopheles aquasalis mosquitoes* were assigned to one of two adult nutritional regimes: either a 10% sucrose solution or a 10% honey solution provided *ad libitum*. After rearing both female and male *Anopheles aquasalis* until the pupa stage, under the same conditions described in the Mosquito Rearing Section, female pupae and male pupae were transferred to plastic cups containing a 0.33% salt solution and placed in paperboard cages (10 cm × 15 cm diameter), with access to either a 10% sucrose solution or a 10% honey solution, both provided *ad libitum*. Females and males were housed together in the same cage throughout the experiment. Seven days after emerging, adult females were exposed to a mouse (Swiss mice lineage—FIOCRUZ CEUA license LW 10/2026) for a blood meal lasting 30 min. Engorged mosquitoes were separated from unfed ones on chilled Petri dishes after cold anesthesia at 4 °C for 5 min. Engorged females were randomly selected from each nutritional diet and placed individually in small paperboard cages (5.5 cm height × 9 cm diameter) for the fecundity and fertility assays. In these small paperboard cages, a diet of either a 10% sucrose solution or a 10% honey solution was provided *ad libitum*, matching the diets before the female blood meals. Two days after the blood meal, plastic cups with a 0.33% salt solution and a filter paper were placed inside each cage to allow for oviposition. Seven days after the first blood meal, another blood meal was administered, and this procedure was repeated until all female mosquitoes had deceased (a total of 7 blood meals were performed). The second and subsequent blood feedings were conducted in the same manner as the first blood feeding. Independent of whether the females fed or not, all females were kept for the entire duration of the experiment, and survival was monitored daily. Blood feeding was performed every seven days until the last mosquito succumbed. The mortality of mosquitoes (both females and males) was monitored daily, and the longevity of each individual was recorded as the number of days from emergence to death. Deceased mosquitoes were promptly removed from the cages each day to prevent any potential influence on subsequent observations. Fecundity, representing the reproductive capacity of female mosquitoes, was assessed by quantifying the total number of eggs produced in each cage over the duration of the seven oviposition events associated with the seven blood meals. The eggs collected from the cages were carefully counted and stored under appropriate conditions for a period of seven days to maintain viability. Following the storage period, the collected eggs were transferred to trays containing water to facilitate hatching into larvae. The trays were maintained under suitable environmental conditions to support larval development and ensure optimal hatch rates for both groups. This process allowed for the evaluation of fertility, as indicated by the total number of hatched larvae from the female mosquitoes.

### 2.5. Statistical Analyses

Statistical analyses were conducted using the software R v4.2.3 (www.r-project.org) assessed on 25 May 2023, and a *p*-value of less than 0.05 was considered statistically significant. The survival data for two groups of mosquitoes, one fed with sugar and the other with honey, were analyzed using Kaplan–Meier survival curves and the log-rank test. The Kaplan–Meier method was employed to estimate the survival functions for both groups, providing a non-parametric statistic that allows for the estimation of survival probabilities over time while accounting for censored data. Survival curves were plotted for each group to visually compare the survival distributions. To statistically compare the survival distributions between the two groups, the log-rank test was performed. This test assesses the null hypothesis that there is no difference in survival between the groups. The log-rank test statistic was calculated, and the corresponding *p*-value was obtained to determine the significance of the observed differences in survival. Fecundity data (number of laid eggs) and fertility data (number of hatched eggs) were analyzed using the Mann–Whitney U test, a non-parametric test used to compare differences between two independent groups. This test was chosen due to its suitability for non-normally distributed data. The Mann–Whitney U test was applied to compare the fecundity and fertility between the two dietary groups (sugar versus honey). The test statistic and the corresponding *p*-values were calculated to determine the significance of the observed differences in fecundity and fertility.

## 3. Results

In the present study, we assessed the survival outcomes of two groups of *Anopheles aquasalis* mosquitoes subjected to different nutritional regimens: one group was provided a sucrose-based diet, while the other received a honey-based diet. During the study, we carefully recorded the temporal progression of mosquito mortality. Additionally, we evaluated the effects of these diets on the longevity, fertility, and fecundity of *Anopheles aquasalis*. Blood-feeding events were conducted at 7-day intervals, allowing time for egg maturation between each cycle, and oviposition was monitored within 4 days post-feeding. This approach enabled us to assess reproductive success across multiple gonotrophic cycles, ensuring the accurate attribution of eggs to the respective feeding events.

### 3.1. Dietary Influence on Survival of Anopheles aquasalis Mosquitoes

To study the effect of diet on the survival of adult *Anopheles aquasalis* mosquitoes, we recorded mosquito mortality over time and performed survival analyses separately for female and male mosquitoes. Kaplan–Meier curves for each group (Figure 1A,B, respectively) were plotted to visualize cumulative survival. We used the log-rank test to assess survival differences between the sucrose and honey treatments, revealing a statistically significant disparity for both female and male *Anopheles aquasalis* mosquitoes. Specifically, the sugar-fed mosquitoes exhibited markedly reduced survival compared to those fed honey (*p* < 0.001 for both females and males). The log-rank test, a robust non-parametric method, was chosen for its ability to compare survival distributions without the proportional hazards assumption required by other methods. The analysis, conducted using the “survdiff” function in R, showed a stark difference in survival outcomes between the two diets.

The mean post-blood-feeding longevity of *Anopheles aquasalis* females in the honey-fed group was 26.1 ± 2.63 SE days, while those in the sucrose-fed group exhibited a significantly lower longevity of 12.1 ± 1.25 SE days (Figure 1). For the female mosquitoes in the honey treatment group, there was an expected number of 130.9 events. The values for (O−E)^2/E and (O−E)^2/V were calculated to be 12.8 and 63.2, respectively. Conversely, in the sugar treatment group, all 90 participants also experienced the event, with an expected number of 49.1 events. The corresponding values for (O−E)^2/E and (O−E)^2/V were 34.0 and 63.2, respectively (Table 1). The chi-square statistic for comparing the two groups was 63.2, with 1 degree of freedom. This yielded a highly significant *p*-value of 2 × 10^−15^, indicating a statistically significant difference in survival times between the honey and sugar treatment groups among female subjects.

In the experiment on males, those in the honey-fed group had a mean post-blood-feeding longevity of 19.2 ± 1.69 SE days, which was significantly higher compared to the 14.8 ± 1.47 SE days observed in the sucrose-fed group (Figure 1). The calculated values for (O−E)^2/E and (O−E)^2/V were 7.38 and 27.9, respectively. Similarly, in the sugar treatment group, all 90 subjects experienced the event, with an expected number of 60.3 events. The values for (O−E)^2/E and (O−E)^2/V were 14.66 and 27.9, respectively (Table 1). The chi-square statistic for the comparison between the two groups was 27.9, with 1 degree of freedom. This resulted in a highly significant *p*-value of 1 × 10^−7^, indicating a statistically significant difference in survival times between the honey and sugar treatment groups. Taken together, these results demonstrate that the type of diet significantly influences the survival of *Anopheles aquasalis* mosquitoes under laboratory conditions. The extended longevity of honey-fed mosquitoes has important biological implications, particularly in the context of malaria transmission dynamics. Longer-lived mosquitoes are more likely to survive long enough to become infectious and transmit malaria parasites to humans, thereby potentially increasing the risk of transmission in natural settings. This highlights the significance of nutritional sources in influencing vector behavior and population dynamics, with far-reaching implications for public health strategies aimed at controlling malaria transmission.

### 3.2. Impact of Adult Diet on Fecundity in Anopheles aquasalis Mosquitoes

To examine the influence of adult diet on the fecundity of *Anopheles aquasalis* mosquitoes, the number of eggs produced under different dietary treatments (sugar vs. honey) was evaluated. Initially, a Shapiro–Wilk normality test was conducted to assess the distribution of the egg count data. The results indicated a significant deviation from normality (W = 0.89093, *p* < 0.001), which justified the use of non-parametric statistical methods for further analysis. A comparative plot was generated to visualize the differences in fecundity between the two treatments (Figure 2). This plot showed the distribution of egg counts, with the honey treatment group exhibiting a noticeably higher median number of eggs compared to the sugar treatment group. Subsequently, a Mann–Whitney U test (also known as the Wilcoxon rank-sum test) was performed to statistically assess the difference in fecundity between the two diets. The analysis revealed a significant difference in egg production, with the honey-fed group producing significantly more eggs than the sugar-fed group (WMann−Whitney U = 6553, *p* < 0.001, *p* = 3.40 × 10^−10^, CI95% [−0.52, −0.30], nobs = 300). This showed that *Anopheles aquasalis* mosquitoes fed with honey exhibited higher fecundity compared to those on the sugar diet.

### 3.3. Dietary Effects on Fertility in Anopheles aquasalis Mosquitoes

To test the impact of adult diet on fertility rates, the total number of hatched eggs in *Anopheles aquasalis* mosquitoes was analyzed using data from different dietary treatments, specifically sugar and honey. To visualize the data, a non-parametric comparison of the total number of hatched eggs between the two dietary treatments was performed (Figure 3 and Appendix A). The Mann–Whitney U test was conducted to statistically compare the total number of hatched eggs between the sugar and honey treatments. The test results showed a significant difference between the two groups (U = 6031, *p* = 2.15487 × 10^−12^). This indicates that this type of diet significantly affects the total number of hatched eggs in *Anopheles aquasalis* mosquitoes. Specifically, the honey treatment resulted in a higher number of hatched eggs compared to the sugar treatment. These findings suggest that the nutritional quality of the adult diet plays a crucial role in the reproductive success of *Anopheles aquasalis* mosquitoes. The higher hatching rates observed in the honey-fed group may be attributed to the richer nutrient profile of honey compared to sugar, which could provide better support for egg development and hatching.

## 4. Discussion and Conclusions

Mosquitoes primarily derive their energy from plant nectars, which also play a crucial role in synthesizing essential molecules like amino acids, nucleotides, and storage compounds such as triglycerides, trehalose, and glycogen [26]. Previous research on *Anopheles gambiae* has demonstrated that plant feeding is vital for their survival, with studies indicating that mosquitoes actively select their feeding sources based on nutrient availability and plant species [30,31]. Furthermore, experimental studies have shown that *An. gambiae* discriminates among plant species, perching and feeding more often on certain species, regardless of whether the plants are presented individually or mixed with others [32].

Recent studies have also demonstrated that the type of plant nectar consumed influences microbiome composition in *Anopheles* mosquitoes, which in turn affects their susceptibility to *Plasmodium* infection. For instance, research by Wang et al. used metabolomics, RNA sequencing, and reverse genetic analysis to explore how plant nectar affects the infectivity of *Plasmodium* parasites in *Anopheles stephensi* mosquitoes [31]. Their metabolomic analysis revealed that *Plasmodium berghei*-infected mosquitoes exhibited significant reductions in trehalose, glucose, succinate, and citrate levels, along with increases in pyruvate and acetate, indicating heightened glycolytic activity due to infection. Mosquitoes fed on trehalose or glucose diets for five days before *Plasmodium berghei* infection showed higher oocyst counts compared to those on a sucrose diet. Notably, *P. falciparum* oocyst numbers increased in mosquitoes fed on trehalose but not on glucose or sucrose [29].

These studies show that diet composition affects the survival of *An. gambiae* and that this species actively chooses where to feed. They also indicate that diet can impact the ability to transmit the parasite through the microbiome of *An. stephensi* mosquitoes. However, the effects of adult diet composition on the longevity and reproductive success of *An. aquasalis* mosquitoes remain poorly studied. Due to its significant public health importance and the ease of laboratory rearing, *An. aquasalis* has been extensively used as a neotropical anopheline model to study host–parasite interactions. Therefore, it is of notable importance to study the dietary effects on adult *An. aquasalis* under laboratory conditions. Additionally, the influence of diet on the gut microbiome cannot be overstated. The recent literature suggests that dietary sources can significantly alter the gut microbiome of mosquitoes, affecting their vector competence and physiological responses. For example, honey, with its diverse nutrient profile, may promote a more beneficial microbiome that enhances mosquito health and resilience, whereas a sucrose-based diet may lead to a less diverse and less protective gut flora [34,35,36,37]. Studies indicate that the microbiota composition in *Aedes aegypti* varies with dietary intake, impacting the mosquito’s susceptibility to dengue and other pathogens [38,39].

Sugar solutions used in laboratory studies typically consist solely of sucrose, a disaccharide that serves as a simple carbohydrate source. In contrast, honey contains a complex mixture of carbohydrates, primarily fructose and glucose, along with small amounts of sucrose. Additionally, honey is rich in other bioactive components, including amino acids, vitamins, minerals, and phytochemicals such as flavonoids and phenolic acids, which are absent in plain sugar solutions. These additional components in honey may provide enhanced nutritional benefits that could explain its observed positive effects on mosquito longevity and fecundity. For example, the antioxidants in honey may reduce oxidative stress, while its micronutrients may support physiological processes that are critical for reproduction. This comparison highlights the broader nutritional profile of honey and its potential to mimic natural nectar and pollen feeding more closely than sucrose solutions.

Our findings indicate that diet type significantly influences mosquito survival and reproductive output. Mosquitoes fed on honey exhibited a markedly longer lifespan and higher fecundity compared to those provided with sucrose. Furthermore, the hatching success rates of eggs from honey-fed females were significantly higher than those from sucrose-fed females. These results highlight the substantial impact of dietary choices on the reproductive fitness of *An. aquasalis*, with critical implications for laboratory studies on vector–parasite interactions. In addition to exploring the microbiome’s role, future studies should also assess how different diets influence susceptibility to various pathogens, such as *Plasmodium* and arboviruses. Research into how sugar feeding interacts with blood feeding to affect reproductive success could broaden the ecological and practical implications of dietary influences. Understanding these dynamics will be essential for improving control strategies targeting mosquito vectors.

In conclusion, our study emphasizes the necessity of carefully considering diet in An. aquasalis laboratory research to ensure accurate assessments. Additionally, broader comparisons with sugar- and nectar-feeding studies across a wider range of mosquito species, such as *Aedes aegypti* and *Culex quinquefasciatus*, would enhance our understanding of dietary influences on survival and reproductive success. Research has shown that nectar feeding can significantly enhance longevity and reproductive success across different species, demonstrating the importance of dietary composition in vector ecology and control [30,31,32]. Nevertheless, to gain a better understanding of the impact of diet on vector–parasite interactions and the vector competence of *An. aquasalis* for *Plasmodium* parasites, future comprehensive studies are crucial. These studies should assess how different diets influence microbiome composition in *Anopheles* mosquitoes and whether this affects their susceptibility to *Plasmodium* infection. This information is vital for improving laboratory study approaches and assessments to obtain insights that closely resemble natural conditions.

## Figures and Tables

**Figure 1 insects-15-00978-f001:**
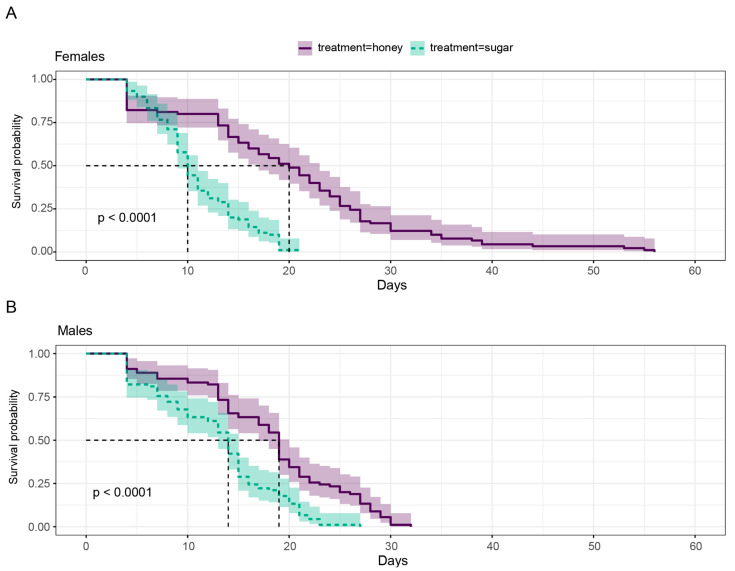
The Kaplan–Meier survival curves and estimates for *Anopheles aquasalis* mosquitoes. Survival probability of *Anopheles aquasalis* females (**A**) and males (**B**). The violet dashed line with light violet shading indicates the honey treatment, while the green solid line with light green shading represents the sugar treatment. Survival distributions were significantly different between the diet exposure groups within females and males (females’ log-rank statistic, df = 2, *p* < 0.0001; males’ log-rank statistic, df = 2, *p* < 0.0001).

**Figure 2 insects-15-00978-f002:**
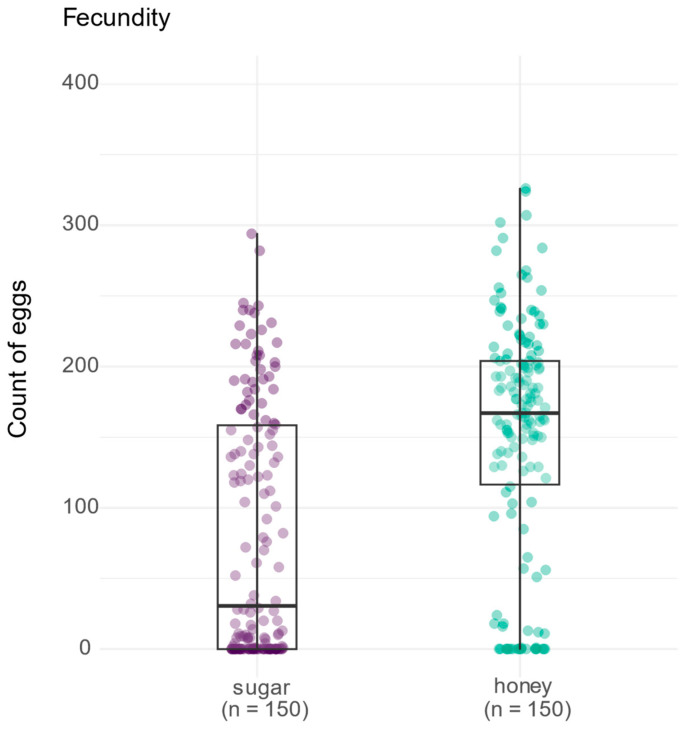
Effect of adult diet on fecundity in *Anopheles aquasalis* mosquitoes. The figure shows the distribution of egg counts for the two diet groups tested: sugar (represented by purple points) and honey (represented by green points). Each individual point represents the egg count from a single female. The boxes in the plot indicate the median and interquartile range for each group, with whiskers showing the data range, excluding outliers. The results of the Mann–Whitney U test indicate a significant difference in egg production between the sugar and honey treatments (W_Mann−Whitney_ = 6553.00, *p* = 3.40 × 10^−10^, rank-biserial r = −0.42, CI_95%_ [−0.52, −0.30], n_obs_ = 300).

**Figure 3 insects-15-00978-f003:**
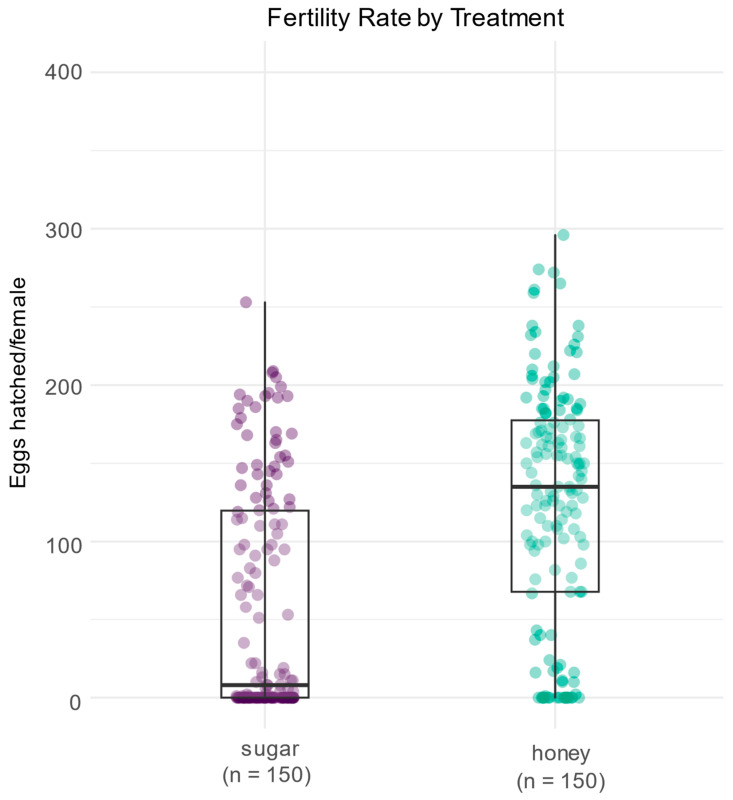
Effect of adult diet on fertility in *Anopheles aquasalis* mosquitoes. The figure shows the distribution of hatched eggs for the two diet groups tested: sugar (represented by purple points) and honey (represented by green points). Each individual point represents the eggs hatched from a single female. The boxes in the plot indicate the median and interquartile range for each group, with whiskers showing the data range, excluding outliers. The results of the Mann–Whitney U test indicate a significant difference in hatched eggs between the sugar and honey treatments (W_Mann−Whitney_ = 6031.00, *p* = 2.15 × 10^−12^, rank-biserial r = −0.46, CI_95%_ [−0.56, −0.36], n_obs_ = 300).

**Table 1 insects-15-00978-t001:** Log-rank test comparison of honey vs. sugar treatments.

**Female**					
**Treatment**	**N**	**Observed**	**Expected**	**(O-E)^2/E**	**(O-E)^2/V**
Honey	90	90	130.9	12.8	63.2
Sugar	90	90	49.1	34.0	63.2
**Male**					
**Treatment**	**N**	**Observed**	**Expected**	**(O-E)^2/E**	**(O-E)^2/V**
Honey	90	90	119.7	7.38	27.9
Sugar	90	90	60.3	14.66	27.9

Female: Chisq = 63.2 with 1 degree of freedom, *p* = 2 × 10^−15^; Male: Chisq = 26.9 with 1 degree of freedom, *p* = 1 × 10^−17^.

## Data Availability

The data presented in this study are openly available in FigShare https://dx.doi.org/10.6084/m9.figshare.27074302 accessed on 9 September 2024.

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
