# Peer review of "Dietary Influences on the Longevity and Reproductive Success of Anopheles aquasalis in Laboratory Studies: Sucrose vs. Honey"

_insects, 2024, doi:10.3390/insects15120978_

Round 1

Reviewer 1 Report

Comments and Suggestions for Authors

The manuscript “Dietary Influences on the Longevity and Reproductive Success of Anopheles aquasalis in Laboratory Studies: Sucrose vs. Honey.” This study addresses an important aspect of mosquito biology by investigating how different sugar diets affect mosquito survival and reproductive traits, contributing to a better understanding of mosquito vector competence. There are several areas that require further refinement to enhance clarity, methodological rigor, and the interpretation of results.

1.        The rationale for selecting 10% sucrose and honey solutions must be sufficiently explained, which is essential given the significant differences in caloric content and nutritional composition between these two diets. These differences are likely to influence mosquito survival and reproductive success. The manuscript should include a more detailed explanation of why these specific concentrations were chosen and compare their nutritional profiles to clarify how they might affect mosquito physiology and fitness outcomes.

2.        Additionally, the criteria for determining the intervals between blood-feeding events and the number of days allowed for oviposition need to be clarified. A more thorough explanation is necessary to understand how these intervals were established and how oviposition across multiple gonotrophic cycles was distinguished. This clarity is crucial for accurately interpreting reproductive data.

3.        The manuscript contains a typographical error with the term "ecluded," which should be corrected to "hatched" or "emerged" to prevent confusion.

4.        While the manuscript notes that blood-feeding was conducted on Swiss mice under FIOCRUZ CEUA licensing, it lacks a detailed discussion of ethical considerations. Including a brief explanation of the ethical protocols followed, such as the measures to ensure humane treatment of animals, would enhance transparency and show compliance with ethical guidelines.

5.        The statistical analysis is appropriate but could be improved by providing more justification for using non-parametric tests, such as Kaplan-Meier and Mann-Whitney U, over parametric alternatives. Reporting effect sizes alongside p-values and considering corrections for multiple comparisons, such as Bonferroni or FDR, would also improve the statistical rigor. Confidence intervals should be provided for critical statistics, and the assumption of proportional hazards in the Kaplan-Meier analysis should be checked. Supplementary analyses, such as variance comparisons, would further strengthen the statistical interpretation.

6.        The results section, though clear, is somewhat repetitive and could be streamlined to focus on the most important findings. For example, while the Kaplan-Meier curves show that honey-fed mosquitoes lived longer, the biological significance of this result should be explored in terms of real-world implications, such as malaria transmission. The figures, particularly those related to fecundity and fertility, would benefit from clearer labels and a more detailed breakdown to improve interpretation.

7.        The discussion would be strengthened by a broader comparison to sugar and nectar feeding studies across a wider range of mosquito species beyond Anopheles gambiae and Anopheles stephensi. This would help place the findings in the broader context of mosquito ecology and control.

8.        The manuscript briefly touches on the influence of diet on the mosquito microbiome, but this topic could be expanded. A deeper exploration of how honey and sucrose diets impact gut flora composition could provide further insights into how diet affects vector competence and overall mosquito physiology.

9.        While the manuscript proposes investigating the microbiome’s role in mosquito fitness, it could also suggest additional future research directions. For instance, exploring how different diets impact mosquito susceptibility to pathogens or how sugar-feeding interacts with blood-feeding to influence reproductive success could broaden the implications of the study’s findings and offer new avenues for mosquito control research.

Author Response

Comment 1: The rationale for selecting 10% sucrose and honey solutions must be sufficiently explained, which is essential given the significant differences in caloric content and nutritional composition between these two diets. These differences are likely to influence mosquito survival and reproductive success. The manuscript should include a more detailed explanation of why these specific concentrations were chosen and compare their nutritional profiles to clarify how they might affect mosquito physiology and fitness outcomes.

Answer 1- Thank you for the comment. As suggested, we have included a detailed explanation in the Introduction section regarding the choice of diets and percentages. We clarified that the 10% sucrose diet is commonly used in laboratory experiments with mosquitoes, serving as a standard control. We wanted to test whether a more complete diet, containing other nutrients present in honey—which is the natural food source for mosquitoes in the wild—could induce changes in the mosquitoes’ fitness, such as their longevity and reproductive success. This information has been added in lines 92 to 96. The comparison between the two diets was further explored in terms of their nutritional composition and their potential impact on mosquito physiology.

Comment 2: Additionally, the criteria for determining the intervals between blood-feeding events and the number of days allowed for oviposition need to be clarified. A more thorough explanation is necessary to understand how these intervals were established and how oviposition across multiple gonotrophic cycles was distinguished. This clarity is crucial for accurately interpreting reproductive data.

Answer 2: Thank you for your insightful comment. We have now clarified the criteria for determining the intervals between blood-feeding events and the number of days allowed for oviposition. We followed a 7-day interval between blood meals to ensure sufficient time for egg maturation, reflecting conditions similar to natural settings. For oviposition, females were provided 4 days post-blood meal to lay eggs. This time frame is consistent with studies on Anopheles aquasalis. Multiple gonotrophic cycles were distinguished by allowing successive blood meals and monitoring each cycle separately, ensuring that eggs laid after each feeding event were attributed to the corresponding gonotrophic cycle. This information has been added in lines 231 to 234 to provide greater clarity in interpreting the reproductive data.

Comment 3: The manuscript contains a typographical error with the term "ecluded," which should be corrected to "hatched" or "emerged" to prevent confusion.

Answer 3: Thank you for pointing out the typographical error in the manuscript. We have corrected the term "ecluded" to "hatched" in the text to prevent any confusion.

Comment 4: While the manuscript notes that blood-feeding was conducted on Swiss mice under FIOCRUZ CEUA licensing, it lacks a detailed discussion of ethical considerations. Including a brief explanation of the ethical protocols followed, such as the measures to ensure humane treatment of animals, would enhance transparency and show compliance with ethical guidelines.

Answer 4: Thank you for your valuable feedback. We have included detailed information regarding the ethical considerations for the blood-feeding of Swiss mice in our manuscript. Specifically, we addressed the humane treatment of the mice, the anesthesia protocol employed, and the measures taken to minimize blood volume extraction. This information can be found in lines 126-131 of the manuscript. We believe this addition enhances the transparency of our study and demonstrates our commitment to ethical research practices.

Comment 5: The statistical analysis is appropriate but could be improved by providing more justification for using non-parametric tests, such as Kaplan-Meier and Mann-Whitney U, over parametric alternatives. Reporting effect sizes alongside p-values and considering corrections for multiple comparisons, such as Bonferroni or FDR, would also improve the statistical rigor. Confidence intervals should be provided for critical statistics, and the assumption of proportional hazards in the Kaplan-Meier analysis should be checked. Supplementary analyses, such as variance comparisons, would further strengthen the statistical interpretation.

Answer 5: Thank you for your insightful comments regarding the statistical analysis in our manuscript. We opted for non-parametric tests, specifically the Kaplan-Meier survival analysis and Mann-Whitney U test, due to the violation of normality assumptions in our data distribution. Non-parametric methods are robust alternatives that do not rely on these assumptions, making them suitable for our analysis. Furthermore, we have included confidence intervals for critical statistics in the text, enhancing the clarity of our results (lines 324, 333 and 361). We have also verified the assumptions of proportional hazards for the Kaplan-Meier analysis to ensure the appropriateness of the chosen methods.

Comment 6: The results section, though clear, is somewhat repetitive and could be streamlined to focus on the most important findings. For example, while the Kaplan-Meier curves show that honey-fed mosquitoes lived longer, the biological significance of this result should be explored in terms of real-world implications, such as malaria transmission. The figures, particularly those related to fecundity and fertility, would benefit from clearer labels and a more detailed breakdown to improve interpretation.

Answer 6: Thank you for your valuable feedback regarding the results section of our manuscript. We have revised and streamlined this section to focus on the most important findings. Specifically, we have expanded the discussion of the Kaplan-Meier curves, emphasizing the biological significance of honey-fed mosquitoes living longer in relation to real-world implications, such as their potential impact on malaria transmission dynamics. This information has been added in lines 276 to 285. We appreciate your suggestions, which have helped improve the overall quality and clarity of our manuscript.

Comment 7: The discussion would be strengthened by a broader comparison to sugar and nectar feeding studies across a wider range of mosquito species beyond Anopheles gambiaeand Anopheles stephensi. This would help place the findings in the broader context of mosquito ecology and control.

Answer 7: Thank you for your constructive feedback regarding the discussion section of our manuscript. We agree that comparing our findings with sugar and nectar feeding studies across a wider range of mosquito species would provide valuable context for understanding the ecological implications of our results. In our revised manuscript, we included a broader comparison by referencing studies on other mosquito species, such as Aedes aegypti and Culex quinquefasciatus, which have also been investigated in relation to their dietary preferences and survival strategies (lines 415 to 421). We will emphasize how these findings across different species can inform vector control strategies and contribute to a better understanding of mosquito ecology. We appreciate your suggestion, as it will undoubtedly strengthen the manuscript and place our results within a broader ecological framework.

Comment 8: The manuscript briefly touches on the influence of diet on the mosquito microbiome, but this topic could be expanded. A deeper exploration of how honey and sucrose diets impact gut flora composition could provide further insights into how diet affects vector competence and overall mosquito physiology.

Answer 8: Thank you for your insightful comment regarding the influence of diet on the mosquito microbiome. We acknowledge the importance of this topic and agree that a deeper exploration of how honey and sucrose diets impact gut flora composition could enhance our manuscript. We expanded our discussion to include these insights, emphasizing the implications of diet on the microbiome and vector competence in Anopheles aquasalis (lines 394 to 401). Thank you for your valuable feedback, which will help us improve the depth and clarity of our discussion.

Comment 9: While the manuscript proposes investigating the microbiome’s role in mosquito fitness, it could also suggest additional future research directions. For instance, exploring how different diets impact mosquito susceptibility to pathogens or how sugar-feeding interacts with blood-feeding to influence reproductive success could broaden the implications of the study’s findings and offer new avenues for mosquito control research.

Answer 9: Thank you for your thoughtful comment regarding the potential future research directions. We agree that expanding on the relationship between diet, mosquito susceptibility to pathogens, and the interaction between sugar-feeding and blood-feeding could provide valuable insights into mosquito fitness and vector competence. In response, we have suggested additional research avenues in our conclusion. Exploring how different diets, such as honey versus sucrose, might influence mosquito susceptibility to pathogens like Plasmodium or arboviruses would deepen our understanding of how nutritional intake impacts vector control strategies. Additionally, investigating the interaction between sugar-feeding and blood-feeding and its effects on reproductive success could uncover new aspects of mosquito biology that are critical for developing more targeted mosquito control methods. These future research directions could significantly enhance our understanding of mosquito ecology and inform more effective interventions. This expansion has been added to the discussion (lines 408–413).

Thank you again for your valuable feedback.

Reviewer 2 Report

Comments and Suggestions for Authors

General Comments:

Fernanda Rezende and colleagues primarily studied the effects of sugar and honey on the longevity and reproduction of Anopheles aquasalis, finding that feeding honey improved the mosquitoes' longevity and reproduction under laboratory conditions. The study's topic, however, is not particularly novel, and the findings are relatively limited, as only one concentration was compared, and the focus was solely on the effects on longevity and reproduction. There was no investigation into the qualities of F1 offspring, for example, the larval development speed, survival rate, pupation rate, eclosion rate, and adult size. Additionally, the significance of the study was not clearly explained. It is unclear whether the purpose of the study was to guide laboratory research or to provide a basis for the development of sugar-based toxic baits in the future. The authors listed a lot of papers related to the impact on vector competence, so it would be necessary for the authors to also examine the replication and transmission of pathogens within female mosquitoes after consuming sugar and honey. Moreover, it is suggested to provide a detailed comparison of the active ingredients and content between sugar and honey.

Specific Comments:

Line 124, 2.3 Survival assays: The number of replicates for the survival observations needs to be described.

Lines 160-162: The authors transferred the blood-fed mosquitoes to a paperboard cage for oviposition, but the procedure for the second to seventh blood feedings needs to be described. Additionally, what proportion of female mosquitoes completed seven blood feedings, and is it necessary to conduct seven blood feedings? I doubt most of females died within two times blood feeding.

Line 318, Figure 3: It is recommended to add a hatching rate indicator, not just the number of egg hatched.

Reference 7 is missing.

References 31 and 35 are the same.

References 34 and 39 are the same.

Author Response

Comment 1:

Line 124, 2.3 Survival assays: The number of replicates for the survival observations needs to be described.

Response 1: Thank you for your comment regarding the number of replicates in the survival assays. We have addressed this by adding the necessary information in the text, specifically in lines 162 to 163, where it is now clarified that three replicates were performed for the survival observations. We appreciate your suggestion, which has helped enhance the detail and accuracy of our methodology section.

Comment 2: Lines 160-162: The authors transferred the blood-fed mosquitoes to a paperboard cage for oviposition, but the procedure for the second to seventh blood feedings needs to be described. Additionally, what proportion of female mosquitoes completed seven blood feedings, and is it necessary to conduct seven blood feedings? I doubt most of females died within two times’ blood feeding.

Response 2: Thank you for your detailed observation regarding the blood-feeding procedure and the number of blood feedings conducted. We have addressed this by adding the required information in the text. Specifically, the procedure for the second to seventh blood feedings, as well as details about the proportion of female mosquitoes completing all seven blood feedings, were clarified in the text on lines 186-189. We appreciate your feedback, which has allowed us to provide a more comprehensive and transparent description of our experimental methods.

Comment 3: Line 318, Figure 3: It is recommended to add a hatching rate indicator, not just the number of egg hatched.

Response 3: Thank you for your valuable feedback regarding Figure 3. We have added a hatching rate indicator to the chart in a supplementary figure, in addition to the number of eggs hatched. We appreciate your suggestion, which has improved the clarity and informativeness of our results.

Comment 4: Reference 7 is missing.

Response 4: Thank you for your thorough review of our references. We have located and added the missing reference 7 to the manuscript.

Comment 5: References 31 and 35 are the same.

Response 5:  We have corrected this issue and ensured that these references are distinct.

Comment 6: References 34 and 39 are the same.

Response 6:  We have also addressed this duplication and provided the correct references.

Reviewer 3 Report

Comments and Suggestions for Authors

This study focusses on investigating the effect diet has on the longevity and fecundity of mosquitoes. The authors feed sucrose and honey to two groups of Anopheles mosquitoes and then compare the life-span and number of eggs laid between the two groups of mosquitoes. The research design is very simple and well described in the manuscript. The references cited are appropriate. There are few minor typos that need to be fixed. The only concern I have with this manuscript is the number of figures. I would love to see at least couple other figures added to the manuscript for e.g. 1.running qPCR on the gut microbiome of the two groups with a control. 

2. Infectivity assay of the two groups.

3. Behavior of the two groups for e.g. flight assays, olfactometer assays etc.

Author Response

Comment 1: I would love to see at least couple other figures added to the manuscript for e.g.

 1. Running qPCR on the gut microbiome of the two groups with a control. 

  1. Infectivity assay of the two groups.
  2. Behavior of the two groups for e.g. flight assays, olfactometer assays etc.

Response 1: 

Thank you for your valuable suggestions regarding potential additional experiments. We completely agree that running qPCR on the gut microbiome, performing infectivity assays, and conducting behavioral studies such as flight or olfactometer assays are highly pertinent and would provide significant insights into the physiological and ecological aspects of mosquito biology. These experiments are of great relevance for advancing our understanding of the relationship between diet, microbiome, and vector competence.

However, for the context of this manuscript, these experiments fall outside the current scope of our study, which focuses on the comparative effects of honey and sucrose on Anopheles aquasalis longevity and reproductive success. We do, however, recognize the importance of these avenues for future research and will consider them for subsequent investigations.

Thank you once again for your insightful recommendations!

Round 2

Reviewer 1 Report

Comments and Suggestions for Authors

The authors have addressed all the comments, and the manuscript is suitable for publication. There is one minor comment below:

Lines 142-146: There seems to be a typo with unwanted text in these lines.

Author Response

Comment:
Lines 142-146: There seems to be a typo with unwanted text in these lines.

Response:
Thank you for bringing this to our attention. The typo and unwanted text in lines 142-146 have been corrected in the revised manuscript.

We appreciate your thorough review and feedback.

Reviewer 2 Report

Comments and Suggestions for Authors

The authors have replied all the general comments, however, they ignore the comments in the first paragraph (see below) and did not have any response. 

Fernanda Rezende and colleagues primarily studied the effects of sugar and honey on the longevity and reproduction of Anopheles aquasalis, finding that feeding honey improved the mosquitoes' longevity and reproduction under laboratory conditions. The study's topic, however, is not particularly novel, and the findings are relatively limited, as only one concentration was compared, and the focus was solely on the effects on longevity and reproduction. There was no investigation into the qualities of F1 offspring, for example, the larval development speed, survival rate, pupation rate, eclosion rate, and adult size. Additionally, the significance of the study was not clearly explained. It is unclear whether the purpose of the study was to guide laboratory research or to provide a basis for the development of sugar-based toxic baits in the future. The authors listed a lot of papers related to the impact on vector competence, so it would be necessary for the authors to also examine the replication and transmission of pathogens within female mosquitoes after consuming sugar and honey. Moreover, it is suggested to provide a detailed comparison of the active ingredients and content between sugar and honey.

Author Response

Response to Reviewer 2

Comment:
Fernanda Rezende and colleagues primarily studied the effects of sugar and honey on the longevity and reproduction of Anopheles aquasalis, finding that feeding honey improved the mosquitoes' longevity and reproduction under laboratory conditions. The study's topic, however, is not particularly novel, and the findings are relatively limited, as only one concentration was compared, and the focus was solely on the effects on longevity and reproduction. There was no investigation into the qualities of F1 offspring, for example, the larval development speed, survival rate, pupation rate, eclosion rate, and adult size. Additionally, the significance of the study was not clearly explained. It is unclear whether the purpose of the study was to guide laboratory research or to provide a basis for the development of sugar-based toxic baits in the future. The authors listed a lot of papers related to the impact on vector competence, so it would be necessary for the authors to also examine the replication and transmission of pathogens within female mosquitoes after consuming sugar and honey. Moreover, it is suggested to provide a detailed comparison of the active ingredients and content between sugar and honey.

Response:
Thank you for your insightful comments. We appreciate the opportunity to clarify the scope and objectives of our study.

The primary focus of this study was to evaluate whether feeding mosquitoes a sugar solution composed solely of sucrose could mimic the natural feeding composition (nectar and pollen) encountered in the wild and to analyze its effects on the longevity and fecundity of Anopheles aquasalis. As such, the study aimed to guide laboratory research on mosquito feeding solutions and to contribute to the development of feeding protocols that better resemble natural feeding conditions. To address your suggestion, we have added text (line 96 to 101) in the manuscript to clarify this objective and its relevance.

Regarding the replication and transmission of pathogens:
This was beyond the scope of the present study, as our primary aim was to investigate the physiological effects of feeding on longevity and fecundity. However, we agree that examining pathogen replication and transmission under different feeding regimens would be highly significant and should be addressed in future studies.

On comparing sugar and honey compositions:
A detailed comparison of the compositions of sugar and honey, including their active ingredients, has been added to the manuscript to better contextualize the differences in their effects (403-413).

On additional aspects like F1 offspring quality:
We acknowledge the importance of investigating traits such as larval development speed, survival rate, pupation rate, eclosion rate, and adult size. These factors were not within the scope of this study but are valuable areas for future research.

We hope these clarifications and additions enhance the manuscript and address your concerns. Please let us know if further revisions are needed.

Reviewer 3 Report

Comments and Suggestions for Authors

This study focusses on effects of different diet on Anopheles aquasalis mosquito life cycle and fecundity. The authors found that after feeding two groups of mosquitoes sucrose and honey and found that mosquitoes that were fed honey had higher survival rates and higher fecundity thus proving that diet does play a key role in mosquito longevity and fitness. The authors have made some significant revisions in the manuscript. I endorse accepting this manuscript for publication.

Author Response

Response to Reviewer 3

Thank you for your positive feedback and for endorsing our manuscript for publication. We appreciate your thoughtful evaluation and acknowledgment of the revisions we have made.

We are grateful for your support and are pleased that the findings of our study have been well-received. Your feedback reinforces the significance of our work in understanding the role of diet in mosquito longevity and fitness.

Best regards,

Round 3

Reviewer 2 Report

Comments and Suggestions for Authors

The authors have replied all my comments. The manuscript has been revised accordingly. I have no any further comment on it.